# Risk Factors of Periprosthetic Infection in Patients with Tumor Prostheses Following Resection for Musculoskeletal Tumor of the Lower Limb

**DOI:** 10.3390/jcm9103133

**Published:** 2020-09-28

**Authors:** Toshifumi Fujiwara, Toshihiro Ebihara, Kazuki Kitade, Nokitaka Setsu, Makoto Endo, Keiichiro Iida, Yoshihiro Matsumoto, Tomoya Matsunobu, Yoshinao Oda, Yukihide Iwamoto, Yasuharu Nakashima

**Affiliations:** 1Department of Orthopaedic Surgery, Graduate School of Medical Sciences, Kyushu University, 3-1-1, Maidashi, Higashi-ku, Fukuoka-shi, Fukuoka 812-8582, Japan; newbie.debity@gmail.com (T.E.); kazuki.kitade@gmail.com (K.K.); sets0rockandsnow@gmail.com (N.S.); makendo@ortho.med.kyushu-u.ac.jp (M.E.); iida-k16@ortho.med.kyushu-u.ac.jp (K.I.); ymatsu@ortho.med.kyushu-u.ac.jp (Y.M.); yasunaka@ortho.med.kyushu-u.ac.jp (Y.N.); 2Department of Orthopaedic Surgery, Kyushu Rosai Hospital, Fukuoka 800-0296, Japan; matsunob@ortho.med.kyushu-u.ac.jp (T.M.); yiwamoto@ortho.med.kyushu-u.ac.jp (Y.I.); 3Department of Anatomic Pathology, Pathological Sciences, Graduate School of Medical Sciences, Kyushu University, Fukuoka 812-8582, Japan; oda@surgpath.med.kyushu-u.ac.jp

**Keywords:** lower limb tumor prosthesis, musculoskeletal tumor, periprosthetic infection

## Abstract

Tumor prostheses for the lower limb following resection of musculoskeletal tumors is useful limb salvage management; however, as compared with routine total joint replacement, an increased incidence of deep periprosthetic infection of tumor prosthesis has been observed. The risk factors for periprosthetic infection of tumor prosthesis remain unclear. This study examines the risk factors and outcomes of periprosthetic infection. This was a retrospective observational study including 121 patients (67 males and 54 females) who underwent tumor prosthesis of the lower limb after resection of musculoskeletal tumors between 1 January 2000 and 30 November 2018. Among a total of 121 tumor prostheses, 7 were total femurs, 47 were proximal femurs, 47 were distal femurs, and 20 were proximal tibias. The incidence of postoperative infection and its risk factors were analyzed. Forty-five patients (37%) had osteosarcoma, 36 patients (30%) had bone metastasis, and 10 patients (8%) had soft-tissue tumors invading the bone. The mean operating time was 229 min, and the mean follow-up duration was 5.9 years. Deep periprosthetic infection was noted in 14 patients (12%). In the multivariate analysis, the risk factors for postoperative infection were identified as being male (hazard ratio [HR], 11.2316; *p* = 0.0100), soft-tissue tumor (HR, 52.2443; *p* = 0.0003), long operation (HR, 1.0056; *p* = 0.0184), and radiotherapy (HR, 6.5683; *p* = 0.0476). The incidence of periprosthetic infection in our institution was similar to that of previous reports. Patients undergoing tumor prosthesis of the lower limb who were male, had a soft-tissue tumor, were predicted to have a long operation, and who underwent radiation, had an increased possibility of postoperative infection.

## 1. Introduction

The development of multimodal treatment for musculoskeletal tumors has improved patient survival and limb-sparing procedures. Tumor prosthesis is the standard reconstruction method for patients with extremity tumors after resection of the long bone, as it provides initial stability, quickly restores function, and is associated with a good long-term outcome. Several studies have reported the long-term excellent survival rate of tumor prosthesis [1,2,3,4,5,6]. Nevertheless, additional operations such as debridement and revision of failed implant are associated with complications such as periprosthetic infection, aseptic loosening, mechanical failure, and fracture. Notably, deep infection remains a major complication of total joint replacement, and the incidence of infection in tumor prosthesis (8%–15%) has been shown to be much higher than that in conventional joint prosthesis (1%–2%) [7,8,9,10,11,12,13,14,15]. Periprosthetic infection exposes the patient to multiple repeated surgical interventions, long-term treatment, residual pain, and even amputation. Several risk factors related to periprosthetic infection of the tumor prosthesis have been reported, including radiotherapy, tibial site, previous surgery, and poor soft-tissue cover [9,10,16]. As reported recently, coated prosthetic reconstruction including silver or iodine prevented postoperative infection of the tumor prosthesis [7,17]. Improved knowledge of the risk factors for periprosthetic infection could result in safer reconstruction after tumor resection for each patient.

Previous reports have demonstrated that periprosthetic infection in the upper limb after tumor prosthesis is rare (0%–2.8%) as compared with that in the lower limb (7%–25%) [9,10]. In addition, the lower extremity was the most frequent developmental location of musculoskeletal tumors [18] and hence focusing on the tumor prosthesis of the lower limb was important to elucidate risk factors of periprosthetic infection of tumor prosthesis. Several studies have recommended that two-stage revision is more successful than one-stage revision or debridement for the treatment of an infected prosthesis [9,10,19]. In contrast, the two-stage revision procedure is extremely difficult and results in low bone stock and less covered soft-tissue.

The purpose of this study was to investigate the risk factors and outcomes of periprosthetic infection of the lower limb after tumor resection in our institution.

## 2. Materials and Methods

### 2.1. Patients

In this single-center retrospective study, we reviewed the medical records of all consecutive patients who underwent reconstruction by tumor prosthesis of the lower limb after excision of bone and/or soft-tissue tumors at Kyushu University Hospital between 1 January 2000 and 30 November 2018. The tumor prostheses of the lower limb included total femur (TF), proximal femur (PF), distal femur (DF), and proximal tibia (PT) and included the Kotz Modular Femur and Tibia Reconstruction System, the Howmedica Modular Reconstruction System, the Global Modular Reconstruction System, the Kyocera Limb Salvage, and the Orthopaedic Salvage System. Recorded data included the pathological diagnosis; tumor site; occurrence from bone only or soft-tissue with invasion of bone; treatments including surgery, chemotherapy, and radiotherapy; postoperative complications; and outcomes of tumors. Noninfected reoperation included surgery for mechanical failure such as implant fracture, instability, and periprosthetic fracture; routine maintenance surgery, such as extension of prosthesis and changing bush; and open reduction of joint dislocation. The records of patients with deep periprosthetic infection were reviewed in detail, including the time of infection, organism, treatment, frequency of surgery, and outcome. This retrospective study was approved by the Ethics Committee of Kyushu University Hospital (approval number: 2020-184).

### 2.2. Definition of Infection

This study defined deep periprosthetic infection as the presence of fistula, positive microbiological culture, periprosthetic pus, histologic infective evidence, and/or clinical evidence of infection such as fever and elevated inflammatory markers in a blood test (e.g., white blood cell count, C-reactive protein, erythrocyte sedimentation reaction, or procalcitonin). All infected cases required surgical treatment.

### 2.3. Surgical Procedure

All infected patients were initially treated with a one-stage procedure, which consisted of the removal of implants, debridement of necrotic and infected tissue, and revision of new components. Briefly, all exchangeable components and polyethylene parts, except the anchorage components, were removed. Debridement consisted of the resection of septic necrotic soft tissue and bone, and rinsing with pulsed lavage was performed. After rinsing with saline including povidone-iodine, the new components were replaced. The Hickman catheter, which was commonly utilized for long-term central venous access, and the drain tube were inserted and placed around the components, following closing a layer of periarticular muscle and skin. After undergoing the one-stage procedure, intra-articular high-dose antibiotics were injected everyday by indwelling catheter until inflammation improved [20]. In cases of infection recurrence, a re-one-stage procedure, two-stage procedure, musculocutaneous flap, or amputation was performed according to the patient’s condition, such as poor soft-tissue coverage, frequent reinfection, or tumor recurrence. The two-stage procedure consisted of placing antibiotic-loaded cement followed by revision of the new implants.

### 2.4. Outcome Measurement

Primary outcomes were reoperation for periprosthetic infection and overall tumor survival in each prosthesis. Secondary outcomes were the analysis of risk factors including age at the time of the initial surgery, sex, body mass index (BMI), site, primary or metastatic tumor, surgical history, occurrence from bone only or soft-tissue tumor invading bone, operating time, adjuvant therapy, and complications of diabetes mellitus.

### 2.5. Statistical Analysis

Statistical analysis was performed using JMP pro 14.0.0 (SAS Institute, Cary, NC, USA). In patients with or without infection, categorical variables were compared using Pearson’s chi-squared test, and continuous variables were analyzed by Mann–Whitney *U* test. The occurrences of tumor survival and periprosthetic infection in patients after initial prosthesis were analyzed according to the Kaplan–Meier method. To identify the independent risk factors of periprosthetic infection of the tumor prosthesis, we used Cox proportional hazards. The Lasso approach was used to select the variables for the multivariable analysis from among the statistically significant variables determined by univariable analysis and reported previously. The receiver operating characteristic (ROC) curve was plotted to evaluate the risk of periprosthetic infection in operating time. The optimal cut-off value was found by calculation of corresponding sensitivity, specificity, positive predictive value (PPV), and negative predictive value (NPV). A statistical difference was defined as *p* < 0.05 for all comparisons. Data represent the mean ± standard deviation (median, range).

## 3. Results

### 3.1. Baseline Characteristics of Patients

In total, this study analyzed 121 consecutive patients (male 67, female 54) undergoing resection of tumors and tumor prosthesis of the lower limb at Kyushu University Hospital from 1 January 2000 to 30 November 2018 (Table 1). The mean age at the time of tumor prosthesis operation was 42.1 years (range, 7–84; median, 42). Of the patients, 45 (37%) had osteosarcoma, 36 (30%) had bone metastasis, and 17 (14%) had chondrosarcoma. Bone metastasis occurred significantly around the PF and was treated by PF implant. Ten patients (8%) with soft-tissue tumor underwent tumor prosthesis responsible for invasion to the bone. In terms of prosthesis type, TF was performed in 7 patients, PF in 42 patients, DF in 45 patients, and PF in 20 patients. The average operating time was 229 min (range, 100–856; median, 210), and the mean follow-up duration was 5.9 years (range, 0.1–19.8; median, 4.2). Adjuvant treatment such as chemotherapy, local radiotherapy, or noninfected reoperation was performed in 65 (54%), 11 (9%), and 17 (14%) patients, respectively. Local radiotherapy was performed at both bone (8%) and soft tissue tumor (1%), and there was no statistical association (*p* = 0.9168, Pearson’s chi-squared test). Noninfected reoperation included extension of prosthesis (2%), open reduction for dislocation of femoral head (2%), revision of prosthesis (6%), osteosynthesis for femoral shaft fracture (1%), secondary suture (1%), relaxation incision for compartment syndrome of lower leg (1%), and re-inosculation of musculocutaneous flap for early venous congestion (1%). All noninfected revised prosthesis was performed one time because of instability or aseptic loosening, and no uninfected multiple revision had undergone in our series. Comorbidity of diabetes mellitus was present in 11 patients (9%), and deep periprosthetic infection was noted in 14 patients (12%).

Kaplan–Meier survival curve analysis revealed that the 10-year overall survival after the initial operation of the tumor prosthesis of the lower limb was 14% (TF), 58% (PF), 70% (DF), and 71% (PT), respectively (Figure 1). Patients who underwent TF had the lowest survival rate in the total prostheses of the lower limb (log-rank test, *p* = 0.0364). In particular, only one patient who underwent TF had survived longer than 10 years; the others were dead within 8 years.

### 3.2. Characteristics and Treatment of the Patients with Deep Periprosthetic Infection

As shown by Table 2, a total of 14 patients (PF, 3 patients; DF, 7 patients; PT, 4 patients) were diagnosed with deep periprosthetic infection and received surgical treatment. The 10-year implant survival rate without infection was 100% (TF), 89% (PF), 79% (DF), and 60% (PT) by the Kaplan–Meier survival curve (Figure 2). Among the infected patients, three died from the underlying tumor. Five patients (36%) were in remission after undergoing one or two one-stage operations; on the other hand, three patients required more than three one-stage surgeries, and another three patients received additional treatment, such as secondary revision, amputation, and musculocutaneous flap. One patient underwent amputation above the knee due to tumor recurrence around the popliteal artery. The patient who underwent PT for giant cell tumor of bone required musculocutaneous flap because of skin and soft-tissue defect after infection. Periprosthetic infection within two years was noted in six patients (one PF, four DFs, and one PT); eight patients had infection after two years (two PFs, three DFs, and three PTs).

### 3.3. Risk Factors for Deep Periprosthetic Infection in Tumor Prosthesis of the Lower Leg after Tumor Resection

Table 3 shows the univariate analysis of each factor between the noninfected patients (*n* = 107) and infected patients (*n* = 14). Previous reports [7,9,10,21] have suggested that the risk factors of tumor prosthesis include prolonged operating time, extensive soft-tissue dissection, PT, adjuvant chemotherapy, radiotherapy, previous surgery, and revision surgery; thus, we compared these factors. We found no difference in age, sex, BMI, implant location (PT vs. other), operating time, chemotherapy, radiotherapy, noninfected reoperation, or diabetes mellitus. On the other hand, soft-tissue tumor invading bone (*p* = 0.0033), primary tumor (*p* = 0.0491), and previous surgery (*p* = 0.0033) were associated with a statistically significant increase in the risk of deep periprosthetic infection. Previous surgery consisted of curettage, resection of soft-tissue tumor, and osteosynthesis for femoral fracture. Treatment of the soft-tissue tumor invading bone required extensive resection of the soft-tissue and bone, indicating the possibility that the implant was poorly covered. No patient who underwent revised prosthesis had been included in the infection group.

Finally, univariate and multivariate Cox proportional hazard models were analyzed for the risk factors of periprosthetic deep infection. The risk factors in the multivariate analysis were selected from the factors shown in Table 4 using the Lasso approach. This was performed after the analysis of the risk factors for deep periprosthetic infection from the isolated variables (sex, soft-tissue tumor, location of implant, previous surgery, operating time, noninfected reoperation, chemotherapy, and local radiotherapy) using the multivariate Cox proportional hazards model. Interestingly, the increased risk factors were found to be male sex (hazard ratio [HR], 11.2316; 95% confidence interval [CI], 1.7843–70.7002; *p* = 0.0100), soft-tissue tumor (HR, 52.2443; 95% CI, 6.0707–449.6165; *p* = 0.0003), operating time (HR, 1.0056; 95% CI, 1.0009–1.0103; *p* = 0.0184), and local radiotherapy (HR, 6.5683; 95% CI, 1.0199–442.3017; *p* = 0.0476) (Table 4). ROC curve was plotted to calculate the risk of periprosthetic infection in operating time, and the area under the curve (AUC) was 0.54406 (95% CI, −0.0002–0.0087), following to find the optimal cut-off value as 493 min (Figure 3). The corresponding sensitivity, specificity, PPV, and NPV were 21%, 98%, 60%, and 91%, respectively.

## 4. Discussion

In this study, we retrospectively investigated the risk factors and outcomes of periprosthetic infection of the lower limb after tumor resection. The rate of deep periprosthetic infection was 12%, and the multivariate Cox hazard model demonstrated that the risk factors were being male, soft-tissue tumor, long operation, and radiotherapy.

The development of treatment for bone and soft-tissue tumor, including chemotherapy, improved surgical techniques, and radiotherapy, has improved the overall survival of patients and been able to salvage affected limbs. Although our cohort, except for the TF group, experienced good 10-year overall survival, the rate of 10-year implant survival without infection was decreased relative to the 5-year rate. Periprosthetic infection is a major postoperative complication of prosthesis. Tumor prosthesis has been shown to have a much higher postoperative infection rate (8%–15%) [10,19,21] compared with routine total joint replacements such as total hip arthroplasty and total knee arthroplasty (1%–2%) [22]. Similar to previous studies, deep periprosthetic infection was found and treated in 14 (12%) of 121 patients in our cohort. Treatment of deep periprosthetic infection has been reported as either a one-stage revision or two-stage revision, and two-stage revision has demonstrated a lower incidence of reinfection [10,19,23,24,25,26]. Our series has also shown that only five patients (36%) could be controlled by one-stage revision; the other patients required multiple or additional surgery such as two-stage revision and musculocutaneous flap. Because two-stage revision was more invasive and required the removal of the entire implant, including the anchorage stem, our initial choice of treatment seemed to be one-stage revision, considering the general condition of the patients.

To date, there have been few reports on the risk factors of periprosthetic infection of tumor prosthesis [9,10,21,27]. Among these studies, the incidence of postoperative infection in the endoprosthesis of the humerus was much lower than in the lower extremity. Because the prevention of periprosthetic infection of the lower extremity, a current clinical problem due to the higher incidence of musculoskeletal tumors in the lower extremity, we analyzed the risk factors of infection in the tumor prosthesis of the lower limb. Using multivariate Cox hazard ratio analysis, we found that being male, having a soft-tissue tumor invading the bone, a long operation duration, and the receipt of radiotherapy were risk factors for postoperative infection after tumor prosthesis of the lower limb, as shown by Table 4. In total hip and knee joint arthroplasty, male sex has been shown to be a risk factor of periprosthetic infection [28,29,30,31,32,33], however, the reason for this remains unclear. Willis-Owen et al. proposed that it could be attributable to sex differences in skin colonization associated with skin pH, sebum production, or skin thickness of the lower limb [31]. Moreover, Kong et al. considered that male patients were more active than female patients and might cycle their implant, leading to an increase in the risk of infection [29]. This study has also found being male to be a risk factor, likely because of the analysis of the lower extremity. Soft-tissue tumor invading the bone requires resection of bone and excessive soft-tissue, including multiple muscles, wide skin, vessels, and nerves, resulting in the possibility of inadequate soft-tissue coverage of the prosthesis. Because poor soft-tissue coverage is a risk factor of periprosthetic infection, soft-tissue tumor invading the bone, as in soft-tissue sarcoma in patients undergoing tumor prosthesis, would increase the risk of infection [21]. Our study also found that prolonged operative time was associated with an increased risk of postoperative infection, as previously reported [21,27,29,34]. In particular, each 20-min increase in operating time elevated the risk of periprosthetic infection of routine joint arthroplasty; thus, decreasing the operative time as much as possible is important for preventing infection. ROC curve showed that operative time over 8 h increased the risk of periprosthetic infection. Despite the low number of patients who received radiotherapy (9%), multivariate analysis indicated that local radiotherapy is a risk factor of periprosthetic infection. As compared with patients treated without radiation, radiotherapy has been demonstrated to be associated with a risk of infection after soft-tissue sarcoma resection, owing to the failure of soft-tissue [10,21,35,36]. In the treatment strategy of sarcoma, the timing of radiotherapy has remained controversial, because both preoperative and postoperative radiotherapy increase the risk of wound complications. In our cohort, the PT group did not display risk factors of infection because almost all patients (90%) had undergone gastrocnemius flap, as previously reported [10].

Silver- and iodine-coated implants have exhibited good outcomes in preventing periprosthetic infection [7,17,19,27,37]. As both coated implants could reduce the risk of infection and prevent periprosthetic infection in higher-risk patients, coated implants should be used for patients with high risk factors, including male, soft-tissue sarcoma, predicted prolonged operative time, and radiotherapy.

Our study has several limitations. First, this was a retrospective study analyzing clinical data. Second, this study examined only a small number of patients from a single institution, and there was no control group. However, because the patients who underwent tumor prosthesis were heterogenous in terms of tumor type, location, resected tissue, and reconstruction type, there was no strict control group; in addition, 35 patients ultimately died of their disease.

Our retrospective study demonstrated the risk factors and outcomes of periprosthetic infection of the lower limb after tumor resection and tumor prosthesis. Deep periprosthetic infection was observed in 12% of patients, and the multivariate Cox hazard model indicated that the risk of periprosthetic infection was increased in patients who were male, had soft-tissue tumor, underwent a long operation, and received radiotherapy.

## Figures and Tables

**Figure 1 jcm-09-03133-f001:**
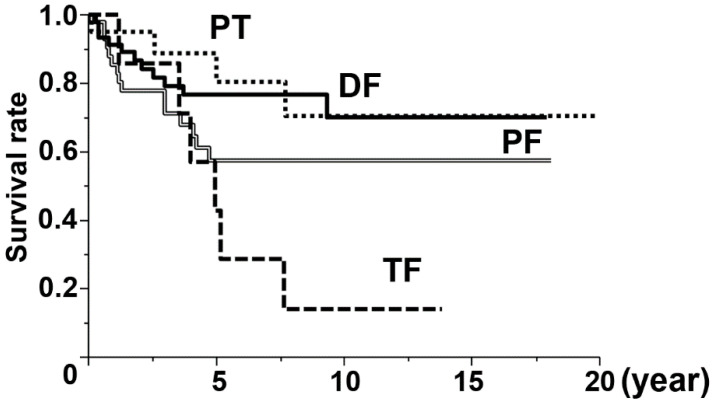
Overall survival of 121 patients who underwent each type of tumor prosthesis. The 5-year and 10-year survival rates were 43% and 14% (Total Femur [TF], long dotted line), 58% and 58% (Proximal Femur [PF], double line), 76% and 70% (Distal Femur [DF], solid line), and 81% and 71% (Proximal Tibia [PT], short dotted line), respectively.

**Figure 2 jcm-09-03133-f002:**
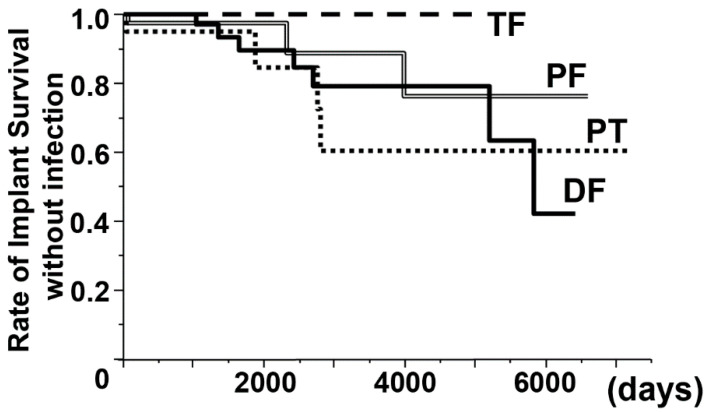
Rate of implant survival without infection. The rates of 5-year and 10-year implant survival without infection were 100% and 100% (Total Femur [TF], long dotted line), 98% and 89% (Proximal Femur [PF], double line), 90% and 79% (Distal Femur [DF], solid line), and 95% and 60% (Proximal Tibia [PT], short dotted line).

**Figure 3 jcm-09-03133-f003:**
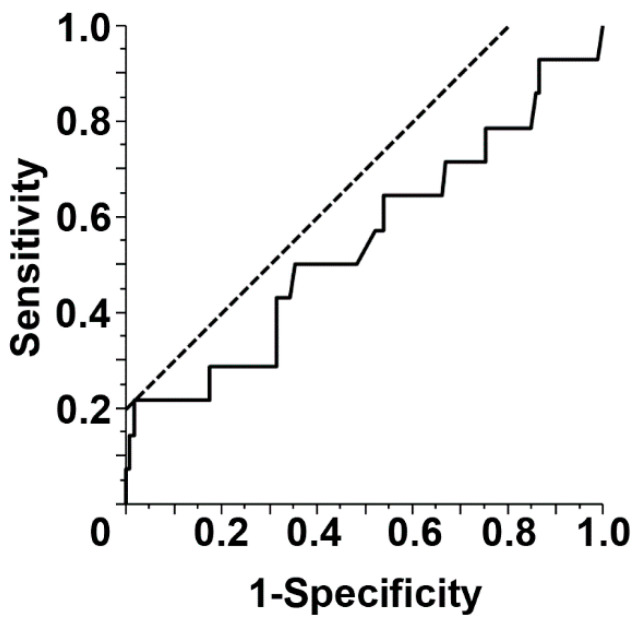
The receiver operating characteristic (ROC) curve for the risk of periprosthetic infection in operating time. The area under the curve (AUC) was 0.54406 (95% CI, −0.0002–0.0087), and the optimal cut-off value was 493 min.

**Table 1 jcm-09-03133-t001:** Demographic data of 121 cases with tumor prosthesis after resection of musculoskeletal tumor of the lower limb.

		Type of Prosthesis			
Parameter	Total	Total Femur	Proximal Femur	Distal Femur	Proximal Tibia
Number	121	7	47	47	20
Age at initial surgery (median, range)	42.1 (42, 7–84)	31.6 (16, 9–73)	53.8 (58, 7–82)	37.3 (26, 8–84)	29.9 (19, 13–74)
Sex					
Male, *n* (%)	67 (55)	2 (29)	26 (55)	26 (55)	10 (50)
Female, *n* (%)	54 (45)	5 (71)	21 (45)	21 (45)	10 (50)
BMI at initial surgery (median, range)	21.6 (21.2, 12.0–32.6)	19.3 (18.8, 12.0–32.6)	22.6 (22.7, 13.7–31.2)	21.4 (20.4, 13.3–24.9)	20.4 (20.1, 13.1–25.2)
Tumor entity, *n* (%)					
Bone tumor	111 (92)	7 (100)	43 (91)	42 (89)	19 (95)
Osteosarcoma	45 (37)	4 (57)	2 (4)	25 (53)	14 (70)
Chondrosarcoma	17 (14)	2 (29)	8 (17)	5 (11)	2 (10)
UPS	3 (2)	0 (0)	2 (4)	1 (2)	0 (0)
Leiomyosarcoma	2 (2)	0 (0)	0 (0)	2 (4)	0 (0)
Ewing sarcoma	1 (1)	1 (14)	0 (0)	0 (0)	0 (0)
Epithelioid angiosarcoma	1 (1)	0 (0)	0 (0)	1 (2)	0 (0)
GCT of bone	6 (5)	0 (0)	0 (0)	4 (9)	2 (10)
Bone metastases	36 (30)	0 (0)	31 (66)	4 (9)	1 (5)
Soft tissue tumor	10 (8)	0 (0)	4 (9)	5 (11)	1 (5)
Synovial sarcoma	3 (2)	0 (0)	2 (4)	1 (2)	0 (0)
UPS	3 (2)	0 (0)	2 (4)	1 (2)	0 (0)
Leiomyosarcoma	2 (2)	0 (0)	0 (0)	1 (2)	1 (5)
ASPS	1 (1)	0 (0)	0 (0)	1 (2)	0 (0)
Diffuse type GCT of TS	1 (1)	0 (0)	0 (0)	1 (2)	0 (0)
Surgery before prosthesis, *n* (%)	10 (8)	0 (0)	3 (6)	5 (11)	2 (10)
Type of implant, *n* (%)					
KMFTR	2 (2)	2 (29)	0 (0)	0 (0)	0 (0)
Growing Kotz	7 (6)	3 (43)	0 (0)	4 (9)	0 (0)
KLS	2 (2)	0 (0)	1 (2)	0 (0)	1 (5)
OSS	1 (1)	0 (0)	1 (2)	0 (0)	0 (0)
HMRS	81 (67)	2 (29)	31 (66)	34 (72)	14 (70)
GMRS	28 (23)	0 (0)	14 (30)	9 (19)	5 (25)
Operative time, minute (median, range)	229 (210, 100–856)	229 (207, 174–330)	213 (191, 100–515)	235 (212, 100–856)	253 (215, 123–511)
Follow-up duration, year (median, range)	5.9 (4.2, 0.1–19.8)	5.8 (4.9, 1.2–13.8)	4.9 (3.0, 0.1–18.1)	6.4 (4.9, 0.3–17.9)	6.6 (5.9, 0.1–19.8)
Oncological status, *n* (%)					
CDF	51 (42)	0 (0)	16 (34)	23 (49)	12 (60)
NED	18 (15)	1 (14)	3 (6)	10 (21)	4 (20)
AWD	16 (13)	0 (0)	13 (28)	3 (6)	0 (0)
DOD	36 (30)	6 (86)	15 (32)	11 (23)	4 (20)
Adjuvant treatments, *n* (%)					
Chemotherapy	65 (54)	5 (71)	18 (38)	28 (68)	14 (70)
Bone tumor	61 (50)	5 (71)	15 (32)	27 (57)	14 (70)
Soft tissue tumor	4 (3)	0 (0)	3 (6)	1 (2)	0 (0)
Local radiotherapy	11 (9)	1 (14)	9 (19)	1 (2)	0 (0)
Bone tumor	10 (8)	1 (14)	8 (17)	1 (2)	0 (0)
Soft tissue tumor	1 (1)	0 (0)	1 (2)	0 (0)	0 (0)
Noninfected reoperation	17 (14)	1 (14)	6 (13)	8 (17)	2 (10)
Extension of prosthesis	2 (2)	0 (0)	0 (0)	2 (4)	0 (0)
Open reduction for dislocation	3 (2)	0 (0)	3 (6)	0 (0)	0 (0)
Revision	7 (6)	0 (0)	2 (4)	4 (9)	1 (5)
Resection of soft tissue tumor	1 (1)	0 (0)	1 (2)	0 (0)	0 (0)
Osteosynthesis	1 (1)	0 (0)	0 (0)	0 (0)	1 (5)
Secondary suture	1 (1)	1 (14)	0 (0)	0 (0)	0 (0)
Relaxation incision	1 (1)	0 (0)	0 (0)	1 (2)	0 (0)
Re-inosculation of musculocutaneous flap	1 (1)	0 (0)	0 (0)	1 (2)	0 (0)
Complication					
Diabetes mellitus, *n* (%)	11 (9)	0 (0)	5 (11)	6 (13)	0(0)
Deep infection, *n* (%)	14 (12)	0 (0)	3 (6)	7 (15)	4 (20)

BMI, body mass index; UPS, undifferentiated pleomorphic sarcoma; ASPS, alveolar soft part sarcoma; GCT, giant cell tumor; TS, tendon sheath; KMFTR, Kotz modular femur and tibia reconstruction system; KLS, Kyocera limb salvage; OSS, orthopedic salvage system; HMRS, Howmedica modular reconstruction system; GMRS, global modular reconstruction system; CDF, continuously disease free; NED, no evidence of disease; AWD, alive with disease; DOD, death on disease.

**Table 2 jcm-09-03133-t002:** Demographics of patients with periprosthetic infection.

No.	Location	Sex	Age	Diagnosis	Bone/Soft Tissue	Implant	Time to Infection, Days	Organism	First Stage, *n*	Second Stage	Status of Tumor	Follow-up Duration, Year
1	PF	M	25	Osteosarcoma	Bone	HMRS	2325	*Staphylococcus epidermidis*	4	No	CDF	15.8
2	PF	M	52	Bone metastasis	Bone	HMRS	3993	*Staphylococcus epidermidis*	4	Secondary Revision	CDF	17.6
3	PF	F	59	Synovial sarcoma	Soft tissue	HMRS	63	*Staphylococcus species*	1	No	DOD	0.6
4	DF	M	19	Osteosarcoma	Bone	HMRS	1033	*Staphylococcus epidermidis*	3	No	CDF	17.9
5	DF	M	20	Osteosarcoma	Bone	HMRS	5825	*Salmonella*	1	No	CDF	16.8
6	DF	M	11	Osteosarcoma	Bone	Growing Kotz	66	*MRSA*	1	Amputation	CDF	6.7
7	DF	M	61	UPS	Bone	HMRS	30	*Staphylococcus lugdunensis*	1	No	CDF	7.4
8	DF	F	21	Synovial sarcoma	Soft tissue	HMRS	55	*Staphylococcus epidermidis*	4	No	CDF	14.3
9	DF	F	73	Leiomyosarcoma	Soft tissue	GMRS	40	*Escherichia coli*	1	No	DOD	3.7
10	DF	M	30	GCT of bone	Bone	HMRS	1621	*Staphylococcus aureus*	2	No	CDF	15.4
11	PT	M	14	Osteosarcoma	Bone	HMRS	1797	*Staphylococcus species*	2	No	DOD	7.7
12	PT	F	74	Leiomyosarcoma	Soft tissue	HMRS	1561	*Unknown*	1	No	CDF	10.4
13	PT	M	40	GCT of bone	Bone	HMRS	1878	*Staphylococcus aureus*	1	Musculocutaneous flap	CDF	7.6
14	PT	M	17	Osteosarcoma	Bone	HMRS	33	*MRSA*	2	No	CDF	9.8

PF, proximal femur; DF, distal femur; PT, proximal tibia.

**Table 3 jcm-09-03133-t003:** Univariate analysis of the factors at baseline.

Variable	Noninfection (*n* = 107)	Infection (*n* = 14)	*p*-Value
Age, y, mean (median, range)	42.8 (44, 7–84)	36.9 (28, 11–74)	0.4985
Sex (male), n (%)	57 (53%)	10 (71%)	0.1987
BMI, mean (median, range)	21.6 (21.2, 12.1–32.7)	21.6 (21.6, 18.5–27.5)	0.7245
Soft-tissue tumor, n (%)	6 (6%)	4 (29%)	0.0033
Location (PT vs. others), n (%)	16 (15%)	4 (29%)	0.1970
Primary (or metastasis), n (%)	72 (67%)	13 (93%)	0.0491
Previous surgery, n (%)	6 (6%)	4 (29%)	0.0033
	Curettage (3), Resection of soft-tissue tumor (2), Osteosynthesis for femoral fracture (1)	Curettage (3), Resection of soft-tissue tumor (1)	
Operative time, min	222.2 ± 83.5 (210, 100–542)	284.9 ± 205.0 (218, 100–856)	0.5928
Chemotherapy, n (%)	58 (54%)	7 (50%)	0.7666
Local radiotherapy, n (%)	9 (8%)	2 (14%)	0.4721
	Bone tumor (8), Soft tissue tumor (1)	Bone tumor (2)	
Noninfected reoperation, n (%)	13 (12%)	4 (29%)	0.0964
	Extension of prosthesis (2), Open reduction for dislocation (1), Revision (7), Resection of soft-tissue tumor (1), Osteosynthesis (1), Secondary suture (1)	Open reduction for dislocation (2), Relaxation incision (1), Re-inosculation of musculocutaneous flap (1)	
Diabetes mellitus, n (%)	10 (9%)	1 (7%)	0.7874
Death at final, n (%)	33 (31%)	3 (21%)	0.4565

Mann—Whitney analysis and Pearson’s chi-squared test. *p* < 0.05 was statistically significant.

**Table 4 jcm-09-03133-t004:** Univariate and multivariate analyses of the risk factors of periprosthetic infection.

	Univariate Analysis			Multivariate Analysis		
	HR	95% CI	*p*-Value	HR	95% CI	*p*-Value
Sex (male)	1.8882	0.5910–6.0323	0.2835	11.2316	1.7843–70.7002	0.0100
Soft-tissue tumor	7.7133	2.3008–25.8587	0.0009	52.2443	6.0707–449.6165	0.0003
Location (PT)	1.5367	0.4788–4.9319	0.4702	3.0272	0.7482–12.2481	0.1204
Previous surgery	7.2761	2.1029–25.1756	0.0017	2.7128	0.6113–12.0396	0.1893
Operative time	1.0055	1.0019–1.0085	0.0006	1.0056	1.0009–1.0103	0.0184
Noninfected-reoperation	0.9290	0.2799–3.0829	0.9042	0.4445	0.1015–1.9473	0.2820
Chemotherapy	0.8250	0.2878–2.3651	0.7204	0.5811	0.1706–1.9793	0.3853
Local radiotherapy	2.1065	0.4583–9.6828	0.3384	6.5683	1.0199–42.3017	0.0476

Variates selected by Lasso approach (age, BMI, primary/metastasis, and DM were excluded). Cox hazard ratio analysis.

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
