# Peer review of "Risk Factors of Periprosthetic Infection in Patients with Tumor Prostheses Following Resection for Musculoskeletal Tumor of the Lower Limb"

_jcm, 2020, doi:10.3390/jcm9103133_

Round 1

Reviewer 1 Report

Good review of periprosthetic joint infection in limb salvage surgery. This is a relatively small number of cases to draw recommendations but this is a rare pathology and therefore 121 patients is acceptable.

I would like the authors to investigate the effect of chemotherapy and multiple revision surgeries (for aseptic reasons) on the risk of infection.

The authors also stated that all patients first underwent a single stage revision before considering a two-stage procedure. There was no mention of DAIR procedures (debridement and implant retention). Can you please clarify this rather important point?

Author Response

I would like the authors to investigate the effect of chemotherapy and multiple revision surgeries (for aseptic reasons) on the risk of infection.

As per the reviewer’s request, we added the detailed chemotherapy and noninfected reoperation in Table 1 and 3, and line 133 of page 9 and line 168 of page10. As shown by page 9, our cohort had no patient undergone multiple revision. We also added the detailed previous surgery before first prosthesis in Table 3 and line 166 of page 10.

Table 4 showed multivariate analysis using Cox hazard model. We selected the risk factors analyzed by multivariate analysis using the Lasso approach. As shown by Table 4, we analyzed the risk factors from sex, soft-tissue tumor, location, previous surgery, operative time, noninfected reoperation, chemotherapy, and radiotherapy. The univariate and multivariate hazard ratio of chemotherapy resulted 0.8250 and 0.5811, respectively.

The authors also stated that all patients first underwent a single stage revision before considering a two-stage procedure. There was no mention of DAIR procedures (debridement and implant retention). Can you please clarify this rather important point?

As reviewer pointed out, we added the detailed one-stage procedure in line 90 of Page 6. Our important point was to rinse with saline including povidone-iodine and to use the Hickman catheter injecting high-dose antibiotics every day until inflammation improved.

Reviewer 2 Report

Thank you for allowing to review the above mentioned paper which contains interesting information regarding risk factors of periprosthetic infection in patients with tumor prostheses although the number is limited with 121 patients. Using multivariate analysis, the risk factors identified were male, soft tissue tumor, lung operation and radiotherapy.

As there were only 8% patients with soft tissue tumors and radiation therapy in 9 % of the cases, it can be assumed that there is an overlap of these two groups, which is expressed also in the high hazard ratio for soft tissue tumors in the rather low one for radiation therapy.  

The authors should go into more details to analyze this point. They also should apply competing risk analysis, which could clarify this obvious overlap.

Where there any Ewing Sarcomas?

Regarding operation time, it would be interesting if the authors could define a Cut Off Value for high and low risk.

Furthermore, there were no tables attached although three tables were mentioned in the manuscript. This could further help to complete the review.

Author Response

As there were only 8% patients with soft tissue tumors and radiation therapy in 9 % of the cases, it can be assumed that there is an overlap of these two groups, which is expressed also in the high hazard ratio for soft tissue tumors in the rather low one for radiation therapy. 

The authors should go into more details to analyze this point. They also should apply competing risk analysis, which could clarify this obvious overlap.

As per the reviewer’s request, we added the date of bone and soft tissue tumor treated with local radiation at Table 1. Local radiotherapy was treated for 10 patients with bone tumor and only one patient with soft tissue tumor. These two factors have shown no statistical association by Pearson’s chi-squared test, as mentioned at line132 of page 9. We also selected risk factors by Lasso approach and analyzed them by multivariate analysis at Table 4, and found radiotherapy and soft tissue tumor invading bone as risk factors of periprosthetic infection.

Where there any Ewing Sarcomas?

Although we have treated many patients with Ewing sarcoma, these patients had been mainly occurred at spine, pelvic, upper limb, fibula, and soft tissue. Therefore, our series had only one patient treated with prosthesis. We discussed in line192 of page12.

Regarding operation time, it would be interesting if the authors could define a Cut Off Value for high and low risk.

As per the reviewer’s suggestion, we calculated the cut-off value of operating time by using ROC curve. The cut-off value of the risk of periprosthetic infection in operating time was 493 minutes, and the corresponding sensitivity, specificity, positive predictive value, and negative predictive value was 21%, 98%, 60%, and 91%, each. We added new figure 3, statistics (line 115 of page 8), result (line 179 of page 11), and discussion (line 229 of page 14).

Furthermore, there were no tables attached although three tables were mentioned in the manuscript. This could further help to complete the review.

We really apologize of not showing table analyzing multivariate analysis. We added the result of multivariate analysis as Table 4. Table 4 showed that the risk factors selected by Lasso approach were analyzed and identified by Cox hazard ratio model.

Round 2

Reviewer 1 Report

Many thanks for revising the article:

Minor changes

  • Please delete the higlighted sentence in lines 213-217. It does not add anything to the reader.
  • Please rephrase the highlighted sentence in lines 251-253: Long operative time over 8 hour was found to be associated with higher risk of developing periprosthetic infection as demonstrated by the ROC curve.

Author Response

Please delete the higlighted sentence in lines 213-217. It does not add anything to the reader.

 Done.

Please rephrase the highlighted sentence in lines 251-253: Long operative time over 8 hours was found to be associated with higher risk of developing periprosthetic infection as demonstrated by the ROC curve.

 As per the reviewer’s suggestion, we have rephrased the sentence, “ROC curve showed that operative time over 8 hours increased the risk of periprosthetic infection.”.